# Optimizing Q-Learning Using Expectile Regression: A Dual Approach to Handle In-Sample and Out-of-Sample Data

## Abstract

Offline Reinforcement Learning (RL) presents unique challenges, primarily due to the constraint of learning from a static dataset without additional environmental interaction. Traditional methods often face limitations in effectively exploiting the available data, particularly when navigating the exploration-exploitation trade-off inherent in RL. This paper introduces a novel algorithm inspired by Implicit Q-Learning Kostrikov et al. (2021), designed to extend the utility of the Bellman update to actions not explicitly present in the dataset. Our approach, termed Extended Implicit Q-Learning (EIQL), strategically incorporates actions beyond the dataset constraints by allowing selection actions with maximum Q. By doing so, it leverages the maximization capability of the Bellman update, while simultaneously mitigating error extrapolation risks. We demonstrate the efficacy of EIQL through a series of experiments that show its improved performance over traditional offline RL algorithms, particularly in environments characterized by sparse rewards or those containing suboptimal and incomplete trajectories. Our results suggest that EIQL enhances the potential of offline RL by utilizing a broader action spectrum.

## 1 Introduction

Reinforcement learning (RL) has made significant advances in solving sequential decision problems in recent years. Training an RL agent typically requires extensive interaction with the environment, which can involve taking actions that are either dangerous or costly. Additionally, online interactions with the environment are time-consuming. Offline RL addresses these challenges by learning from pre-existing datasets without real-time interactions, proving invaluable in practice. However, offline RL must contend with the challenge of distribution shift. Unlike conventional supervised learning, which trains models to perform well on data from the same distribution as the training set, offline RL aims to derive policies that outperform those in the dataset. This involves dynamic programming that might query out-of-distribution (OOD) data, as policies trained on historical data are applied to new and varied state spaces. This can lead to error extrapolation if not properly regularized, underscoring the need for advanced methods capable of adapting to and overcoming these distributional discrepanciesLevine et al. (2020).

To mitigate the distribution shift in offline RL, current approaches involve penalizing OOD state-action pairs or regularizing the trained policy towards the behavior policy. Works such as Kumar et al. (2019) and Kumar et al. (2020) explicitly constrain policy or Q-value updates to ensure pessimism during training. However, balancing the tradeoff between staying within the support of the data and avoiding suboptimal solutions when constraints are too conservative remains challengingKumar et al. (2019). Another approach focuses on eliminating extrapolation error through in-sample learning, which updates the Bellman target using only actions present within the dataset. Implicit Q-Learning (IQL), for instance, employs in-sample learning and avoids querying the value of unseen actions by using expectile regression to assess the near-optimality of actions taken under the observed policy.

In this paper, we introduce a novel algorithm, Extended Implicit Q-Learning (EIQL), which builds on the framework of Implicit Q-Learning. EIQL occasionally allows querying actions not present in

the dataset by leveraging the strengths of Bellman update's maximization capability while mitigating the risk of extrapolation errors by controlling the chance of choosing out-of-sample actions, as demonstrated in our experiments. We test EIQL on the D4RL benchmark, which includes the Gym-MuJoCo locomotion domains and the more challenging AntMaze, Adroit, and Kitchen domains. Our empirical results highlight the effectiveness of EIQL.

## 2 RELATED WORK

Offline reinforcement learning (RL) has recently gained substantial attention due to its potential to learn effective policies from static datasets without additional online exploration. The primary challenge in offline RL is to mitigate extrapolation errors that arise when the learning algorithm queries states and actions that are not well-represented in the dataset. Various approaches have been proposed to address this issue, often focusing on constraining the learned policy to remain within the support of the dataset Levine et al. (2020).

Implicit Q-Learning (IQL) is an offline reinforcement learning algorithm designed to avoid evaluating out-of-distribution actions while still enabling dynamic programming. By employing expectile regression, IQL introduces minor yet effective modifications to the conventional SARSA-like temporal difference learning algorithmKostrikov et al. (2021). Demonstrably, IQL achieves robust performance on the D4RL benchmarkFu et al. (2020).

## 3 PRELIMINARIES

### 3.1 REINFORCEMENT LEARNING

Reinforcement Learning (RL) is a framework where an agent learns to make decisions by interacting with an environment assumed to be a Markov Decision Process (MDP). An MDP is characterized by a set of states $S$, a set of actions $A$, a scalar reward function $R$, transition dynamics $p$, and a discount factor $\gamma$. The agent's interaction with the MDP is governed by a policy $\pi(a|s)$, which can either be a deterministic mapping or a stochastic distribution over actions. The primary objective in RL is to derive a policy that maximizes the expected discounted return, formally expressed as $\mathbb{E}_\pi \left[ \sum_{t=0}^{\infty} \gamma^t r_t \right]$.

### 3.2 OFFLINE REINFORCEMENT LEARNING

In offline reinforcement learning (RL), existing datasets are leveraged to learn policies without the necessity for new data collection. Recent offline RL algorithms often utilize approximate dynamic programming, focusing on minimizing the temporal difference (TD) error. This approach is encapsulated in the loss function $L_{\text{TD}}(\theta)$, defined as:

$$L_{\text{TD}}(\theta) = \mathbb{E}_{(s,a,s') \sim \mathcal{D}} \left[ \left( r(s,a) + \gamma \max_{a'} Q_{\hat{\theta}}(s',a') - Q_\theta(s,a) \right)^2 \right],$$

where $\mathcal{D}$ represents the dataset, $Q_\theta(s,a)$ denotes the parameterized Q-function, and $Q_{\hat{\theta}}(s',a')$ signifies the target network. The policy $\pi(s)$ is derived by maximizing the Q-value, $\pi(s) = \arg\max_a Q_\theta(s,a)$. To mitigate overestimation caused by out-of-distribution (OOD) actions, which can inaccurately elevate $Q_{\hat{\theta}}(s',a')$, modifications to the value function loss and direct constraints on the policy optimization are commonly implemented in recent offline RL strategies.

## 4 EXTENDED IMPLICIT Q-LEARNING

In offline reinforcement learning, leveraging the current Q-network's ability to generalize beyond training data is crucial, yet it can introduce extrapolation errors during loss evaluation. To mitigate this, we propose a value loss function that combines the strengths of Bellman updates and implicit Q-learning for effective sample utilization from the dataset:

$$L_V(\psi) = (1-B) \cdot \mathbb{E}_{(s,a) \sim D} \left[ L_2^\tau \left( Q_{\hat{\theta}}(s,a) - V_\psi(s) \right) \right] + B \cdot \mathbb{E}_{s \sim D, a' \sim \pi(s)} \left[ L_2^{\tau'} \left( Q_{\hat{\theta}}(s,a') - V_\psi(s) \right) \right] \tag{1}$$

Here, $B$ is a Bernoulli random variable with parameter $p$, determining the contribution of each term in the loss:

$$B \sim \text{Bernoulli}(p) \quad \text{where} \quad P(B = x) = \begin{cases} p & \text{if } x = 1 \\ 1 - p & \text{if } x = 0 \end{cases}$$

This stochastic approach allows us to balance the direct Q-values against the maximum expected value from the policy, fostering robust learning.

Additionally, we define the Q-value loss function as:

$$L_Q(\theta) = \mathbb{E}_{(s,a,s') \sim D} \left[ (r(s,a) + \gamma V\psi(s') - Q_\theta(s,a))^2 \right] \tag{2}$$

This equation emphasizes the importance of aligning the estimated Q-values with the rewards and discounted future value, ensuring effective training of the Q-network.

Similar to Implicit Q-Learning (IQL), we employ Advantage Weighted Regression to derive the policy. This objective function trains a policy designed to maximize the Q-value while adhering to a KL divergence constraint between the learned policy, $\pi$, and the sampling policy, $\mu$.

$$\begin{aligned} L_\pi(\phi) = &(1 - B') \cdot \mathbb{E}_{(s,a) \sim D} \left[ \exp \left( \beta \left( Q_{\hat{\theta}}(s,a) - V_\psi(s) \right) \right) \log \pi_\phi(a \mid s) \right] \\ &+ B' \cdot \mathbb{E}_{s \sim D} \left[ \exp \left( \beta \left( Q_{\hat{\theta}}(s,a) - V_\psi(s) \right) \right) \log \pi_\phi(a \mid s) \right] \end{aligned} \tag{3}$$

$$B' \sim \text{Bernoulli}(q) \quad \text{where} \quad P(B' = x) = \begin{cases} q & \text{if } x = 1 \\ 1 - q & \text{if } x = 0 \end{cases}$$

This formulation integrates the exponential weighting of the advantage term, encouraging the policy to focus on actions that provide higher value compared to the current state value, ultimately refining the policy's performance.

### 4.1 EXPECTILE REGRESSION

In our approach, we employ expectile regression to estimate action value functions, both for in-sample and out-of-sample scenarios. This method deviates from traditional approaches that typically rely on the estimation of mean values. We propose that the application of varying expectiles across different gym environments offers distinct advantages.

Particularly, in environments where datasets predominantly feature negative rewards with few positive outcomes, the use of a higher expectile allows for the effective filtration of suboptimal actions. Conversely, in scenarios characterized by datasets containing primarily suboptimal trajectories, employing a lower expectile facilitates a more conservative valuation of action value functions. This methodology aids in mitigating the potential overestimation of Q-values for actions that are poorly represented within the dataset.

---

**Algorithm 1: EIQL**

Initialize parameters $\psi, \theta, \hat{\theta}, \phi$.
for each gradient step do
$\quad \psi \leftarrow \psi - \lambda_V \nabla_\psi L_V(\psi)$
$\quad \theta \leftarrow \theta - \lambda_Q \nabla_\theta L_Q(\theta)$
$\quad \hat{\theta} \leftarrow (1 - \alpha)\hat{\theta} + \alpha\theta$
end for

**Policy extraction (AWR):**
for each gradient step do
$\quad \phi \leftarrow \phi - \lambda_\pi \nabla_\phi L_\pi(\phi)$
end for

---

Our experimental findings, detailed later in this paper, substantiate the efficacy of this approach. By adjusting the expectile level according to the reward distribution characteristics of each dataset, our method not only enhances the robustness of value estimation but also tailors the learning process to accommodate the specific challenges presented by different environments.

### 4.2 BOUNDEDNESS OF VALUE FUNCTION

In the following theorems, we establish that under certain assumptions, our method not only approximates the optimal state-action value function $Q^*$, but also facilitates multi-step dynamic programming.

For clarity and brevity in our analysis, we introduce the following notation: Let $E_\tau^{x \sim X}[x]$ denote the $\tau$-th expectile of a random variable $X$. For example, $E_{0.5}^{x \sim X}[x]$ corresponds to the standard expectation of $X$. We define $V_\tau^I(s)$, the value function in Kostrikov et al. (2021), $V_{\tau,\tau'}(s)$, the value function of our proposed method, and $Q_{\tau,\tau'}(s,a)$ as follows:

$$V_\tau^I(s) = E_{(s,a) \sim D}^\tau[Q_\tau(s,a)],$$

$$V_{\tau,\tau'}(s) = (1-p) \cdot E_{(s,a) \sim D}^\tau[Q_{\tau,\tau'}(s,a)] + p \cdot E_{s \sim D, a \sim \pi(s)}^{\tau'}[Q_{\tau,\tau'}(s,a)],$$

$$Q_{\tau,\tau'}(s,a) = R(s,a) + \gamma E s' \sim P(\cdot|s,a)[V_{\tau,\tau'}(s')],$$

where $\pi(s)$ denotes the policy, $R(s,a)$ is the immediate reward, $\gamma$ is the discount factor, and $P(s'|s,a)$ is the transition probabilityKostrikov et al. (2021). From Kostrikov et al. (2021), we know that $V_\tau^I(s) \leq \max_{\substack{a \in A \\ \pi_\beta(a|s)>0}} Q^*(s,a)$.

**Theorem 4.1** *We assume that there exists an $\epsilon > 0$ such that for all $p \in [0, \epsilon)$,*

$$V_{\tau,\tau'}(s) = (1-p) \cdot E_{(s,a) \sim D}^\tau[Q_{\tau,\tau'}(s,a)] + p \cdot E_{s \sim D, a \sim \pi(s)}^{\tau'}[Q_{\tau,\tau'}(s,a)] \leq \max_{\substack{a \in A \\ \pi_\beta(a|s)>0}} Q^*(s,a),$$

*where $Q^*(s,a)$ is defined as:*

$$Q^*(s,a) = r(s,a) + \gamma \mathbb{E}_{s' \sim p(\cdot|s,a)} \left[ \max_{\substack{a' \in A \\ \pi_\beta(a'|s')>0}} Q^*(s',a') \right].$$

*See Appendix A for detailed proof.*

We have shown that $V_\beta(s)$, the value function under the mixed policy $\pi_\beta$, is bounded by the maximum Q-value achievable under actions taken with positive probability by $\pi_\beta$. This proof highlights the influence of the parameter $\beta$ in controlling the extent to which the new policy deviates from optimality.

### 4.3 LOWER VARIANCE

We start by considering two random variables, $X_1$ and $X_2$, to illustrate the computation of expectations and variances in a Bernoulli distribution framework. Let us define $Y_1$ and $Y_2$ as linear combinations of these variables:

$$Y_1 = (1-p)X_1 + p(X_2)$$

$$Y_2 = (1-B)X_1 + BX_2$$

$$B \sim \text{Bernoulli}(p) \quad \text{where} \quad P(B = x) = \begin{cases} p & \text{if } x = 1 \\ 1-p & \text{if } x = 0 \end{cases}$$

We can easily derive that $\mathbb{E}[Y_1] = \mathbb{E}[Y_2]$:

$$\mathbb{E}[pX_1 + (1-p)X_2] = p\mathbb{E}[X_1] + (1-p)\mathbb{E}[X_2] \tag{4}$$

This equation holds due to the linearity of expectation. We then proceed to calculate the variance of the linear combination:

We then compute the variance of variable $Y_1$ and $Y_2$:

$$\begin{aligned} \text{Var}[Y_1] &= \text{Var}[pX_1 + (1-p)X_2] \\ &= p^2\text{Var}[X_1] + (1-p)^2\text{Var}[X_2] + 2p(1-p)\text{Cov}(X_1, X_2) \end{aligned} \tag{5}$$

$$\text{Var}[Y_2] = p\text{Var}[X_1] + (1-p)\text{Var}[X_2] + p(1-p)\left(E[X_1] - E[X_2]\right)^2 \tag{6}$$

This variance expression arises from the properties of variance and covariance for linear combinations of random variables.

After establishing the general form, we substitute $X_1$ and $X_2$ with the specific expressions involving expectations under different sampling strategies from distribution $D$:

$$X_1 = \mathbb{E}_{(s,a)\sim D}\left[L_2^\tau\left(Q_{\hat{\theta}}(s,a) - V_\psi(s)\right)\right],$$
$$X_2 = \mathbb{E}_{s\sim D,a'\sim\pi(a'|s)}\left[L_2^\tau\left(Q_{\hat{\theta}}(s,a') - V_\psi(s)\right)\right].$$

Applying these substitutions, the variance expressions become:

$$\mathrm{Var}[Y_1] = p^2\mathrm{Var}\left[\mathbb{E}_{(s,a)\sim D}\left[L_2^\tau\left(Q_{\hat{\theta}}(s,a) - V_\psi(s)\right)\right]\right]$$
$$+ (1-p)^2\mathrm{Var}\left[\mathbb{E}_{s\sim D,a'\sim\pi(s)}\left[L_2^\tau\left(Q_{\hat{\theta}}(s,a') - V_\psi(s)\right)\right]\right]$$
$$+ 2p(1-p)\mathrm{Cov}\left(\mathbb{E}_{(s,a)\sim D}\left[L_2^\tau\left(Q_{\hat{\theta}}(s,a) - V_\psi(s)\right)\right], \mathbb{E}_{s\sim D,a'\sim\pi(a'|s)}\left[L_2^\tau\left(Q_{\hat{\theta}}(s,a') - V_\psi(s)\right)\right]\right)$$
$$(7)$$

$$\mathrm{Var}[Y_2] = p\mathrm{Var}[\mathbb{E}_{(s,a)\sim D}\left[L_2^\tau\left(Q_{\hat{\theta}}(s,a) - V_\psi(s)\right)\right]]$$
$$+ (1-p)\mathrm{Var}[\mathbb{E}_{s\sim D,a'\sim\pi(a'|s)}\left[L_2^\tau\left(Q_{\hat{\theta}}(s,a') - V_\psi(s)\right)\right]]$$
$$+ p(1-p)\left(E[\mathbb{E}_{(s,a)\sim D}\left[L_2^\tau\left(Q_{\hat{\theta}}(s,a) - V_\psi(s)\right)\right]] - E[\mathbb{E}_{s\sim D,a'\sim\pi(a'|s)}\left[L_2^\tau\left(Q_{\hat{\theta}}(s,a') - V_\psi(s)\right)\right]]\right)^2$$
$$(8)$$

Given that the policy $\pi(s)$ is designed to generate actions that are supported by the data, the expectations $E[X_1]$ and $E[X_2]$ are expected to be close. The covariance between $X_1$ and $X_2$ is expected to be significantly high. It demonstrate that our approach achieve a reduction in the overall variance of the system. Therefore, our approach to use sampling, as opposed to merely adjusting the weights in the loss function, provides a more effective strategy for minimizing variance.

## 5 EXPERIMENTAL EVALUATION

### 5.1 COMPARISON ON OFFLINE RL BENCHMARKS

We evaluate the performance of EIQL against prior offline RL methods across various domains and dataset compositions, encompassing both continuous and discrete action spaces, as well as state observations with different dimensionalities. Our comparisons include prior offline RL methods: TD3+BCFujimoto & Gu (2021), CQL Kumar et al. (2020), IQLKostrikov et al. (2021), as well as behavioral cloning (BC). The implementations for these algorithms are based on the work of Kostrikov et al. (2021) and Kang et al. (2023).

**Gym locomotion.** We tested our approach using the D4RL benchmark Fu et al. (2020) and compared it to existing methods, as shown in Table 1. Our experiments were conducted in the Gym-MuJoCo locomotion environments, which involve three different agents: halfcheetah, hopper, and walker2d. For each of these agents, five distinct datasets were used. These datasets represent behavior policies of varying quality: random, medium, medium-replay, medium-expert, and expert.

**Adroit tasks.** The Adroit tasks Rajeswaran et al. (2017) in D4RL Fu et al. (2020) involve controlling a 24-DoF robotic hand, which represents a more complex challenge than standard gym tasks. These tasks rely on limited data derived from human demonstrations and are notably difficult due to their complex dataset composition and high dimensionality. The experimental results are shown in Table 2

**Franka kitchen tasks.** The Franka kitchen tasks Gupta et al. (2019) from D4RL Fu et al. (2020), the objective is to control a 9-DoF robot to manipulate various objects, including a microwave and a kettle, within a single episode. The tasks require the robot to achieve a specific configuration for each object, for which it receives a sparse 0-1 completion reward. This domain poses significant challenges due to the need for composing partial trajectories, precise long-horizon manipulation, and the integration of human-provided teleoperation data. These tasks are designed to test the robot's ability to perform complex sequential object manipulation under limited feedback. The experimental results are shown in Table 2

**AntMaze.** The AntMaze taks from D4RLFu et al. (2020) consist of sparse-reward tasks and require leaning more optimal policies from the suboptimal trajectories. EIQL is able to make obvious progress on the more complex large mazes. The results are shown in Table 3.

| Environment | BC | TD3+BC | CQL | IQL | EIQL |
|---|---|---|---|---|---|
| halfcheetah-medium-v2 | 42.6 | **48.1** | 47.0 | 47.5 | 47.7 |
| hopper-medium-v2 | 56.8 | 54.7 | 55.0 | 67.0 | **68.1** |
| walker2d-medium-v2 | 69.5 | 76.2 | 74.5 | 78.3 | **79.9** |
| halfcheetah-expert-v2 | 92.9 | **96.5** | 96.3 | 95.0 | 95.5 |
| hopper-expert-v2 | 107.0 | **110.0** | 96.5 | 108.1 | 109.0 |
| walker2d-expert-v2 | 107.1 | 110.2 | 108.9 | 109.7 | **110.5** |
| halfcheetah-medium-replay-v2 | 36.6 | 43.8 | **45.5** | 44.5 | 45.1 |
| hopper-medium-replay-v2 | 45.1 | 45.5 | 88.7 | 94.5 | **95.1** |
| walker2d-medium-replay-v2 | 23.4 | 42.6 | 81.5 | 73.4 | 75.4 |
| halfcheetah-medium-expert-v2 | 47.1 | **92.4** | 75.7 | 89.3 | 91.1 |
| hopper-medium-expert-v2 | 55.4 | 87.6 | **105.6** | 90.5 | 92.5 |
| walker2d-medium-expert-v2 | 93.3 | 106.5 | **107.9** | 95.7 | 96.1 |

Table 1: Normalized scores on MuJoCo locomotion, averaged over 3 seeds. The implementations of BC, TD3, and CQL are based on Kang et al. (2023)

| Environment | BC | TD3+BC | CQL | IQL | EIQL |
|---|---|---|---|---|---|
| pen-human-v0 | 79.6 | 5.9 | 77.4 | 76.0 | **81.6** |
| pen-cloned-v0 | 33.5 | 17.3 | 40.3 | 45.2 | **58.2** |
| **Total (adroit)** | x | x | x | x | x |
| kitchen-complete-v0 | 64.9 | 2.2 | 42.0 | 65.0 | **68.3** |
| kitchen-partial-v0 | 35.8 | 0.7 | 40.7 | 63.0 | **73.0** |
| kitchen-mixed-v0 | 49.7 | 0.0 | 45.7 | 49.5 | **50.2** |

Table 2: Normalized scores on Adroit and kitchen domains, averaged over 3 seeds. The implementations of BC, TD3, and CQL are based on Kang et al. (2023)

## 5.2 ANALYSIS OF EXPECTILE-BASED VALUE UPDATES

To enhance the robustness of the Bellman update in the value function estimation, we explore the adaptation of expectile regression as an alternative to the conventional mean-based approach. This modification involves fitting the expectile of the return distribution rather than the mean, offering a nuanced view that potentially accommodates the asymmetry in the distribution of returns. We experiment with various levels of the expectile, specifically examining $\tau = 0.5$, which corresponds to the mean-based update, as well as other expectile levels to investigate their impacts on policy performance.

We selected three distinct datasets for our experiments: walker2d-medium-replay-v2, antmaze-large-play-v2, and kitchen-partial-v0 from Fu et al. (2020). These datasets were chosen to represent a diverse array of tasks, complexities, and data distributions, facilitating a comprehensive analysis of expectile-based updates across different domains and scenarios.

Our experiments aim to evaluate how different expectiles influence the learning dynamics and the resultant policy efficacy in offline reinforcement learning settings. By comparing these outcomes, we seek to identify whether expectile regression can provide a more effective or robust approach for value function estimation in scenarios characterized by skewed or heavy-tailed reward distributions.

**antmaze-large-play-v2** In the AntMaze environment, rewards are typically sparse and only given at specific goals or endpoints, such as reaching the end of the maze. Throughout the rest of the maze, the agent might receive zero or even negative rewards for hitting walls or taking inefficient paths. The "large-play" variant features a large maze with minimal guidance on navigationFu et al. (2020). The experimental data presented in the accompanying table demonstrate that higher expectiles enhance learning efficacy for agents by effectively filtering out actions that yield low or negative rewards. As illustrated in Figure 1, higher expectiles are advantageous, particularly in antmaze dataset charac-

| Environment | BC | TD3+BC | CQL | IQL | EIQL |
|---|---|---|---|---|---|
| antmaze-umaze-v2 | 66.5 | 73.0 | 82.7 | 91.0 | **93.3** |
| antmaze-umaze-diverse-v2 | 56.8 | 46.7 | 11.5 | 66.5 | **71.2** |
| antmaze-medium-play-v2 | 0.0 | 0.0 | 59.0 | 74.0 | **78.2** |
| antmaze-medium-diverse-v2 | 0.0 | 0.0 | 53.7 | 73.1 | **75.5** |
| antmaze-large-play-v2 | 0.0 | 0.0 | 15.8 | 44.0 | **61.2** |
| antmaze-large-diverse-v2 | 0.0 | 0.0 | 14.9 | 46.5 | **57.5** |

Table 3: Normalized scores on AntMaze, averaged over 3 seeds. The implementations of BC, TD3, and CQL are based on Kang et al. (2023)

terized by sparse rewards. By excluding actions that lead to negative outcomes, higher expectiles facilitate the development of more effective policies.

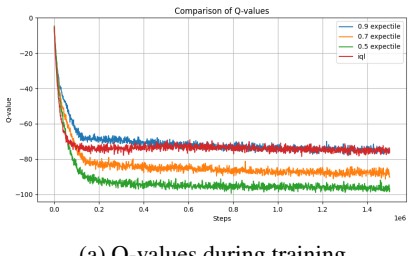 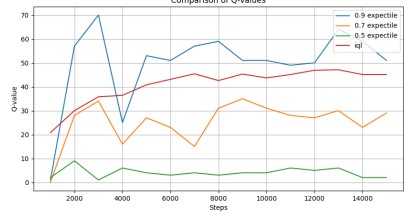

(a) Q-values during training.      (b) Average Evaluation Rewards.

Figure 1: Analysis of the training dynamics and performance evaluation on the antmaze-large-play-v2 dataset: (a) Q-value progression during training across different expectile, (b) Average evaluation rewards across various expectiles.

**kitchen-partial-v0** This dataset features a simulated kitchen environment where an agent interacts with objects. The "partial" variant, characterized by suboptimal actions and incomplete tasksGupta et al. (2019), is analyzed in Figure 2. The results indicate that a lower expectile is advantageous in the kitchen-partial-v0 scenario. It supports the learning of conservative Q-values, thereby mitigating the overestimation of actions associated with suboptimal or incomplete trajectories.

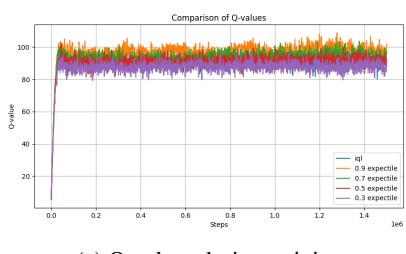 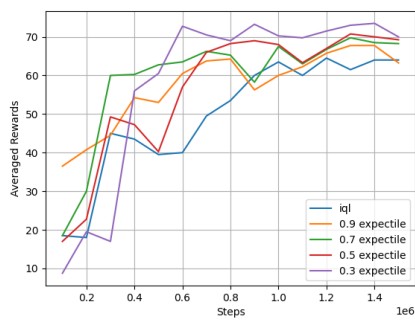

(a) Q-values during training.      (b) Average Evaluation Rewards.

Figure 2: Analysis of the training dynamics and performance evaluation on the kitchen-partial-v0 dataset: (a) Q-value progression during training across different expectile, (b) Average evaluation rewards across various expectiles.

## 6 Conclusion

In this paper, we introduced EIQL, a novel algorithm that engages in both in-sample and out-of-sample learning. This approach leverages the stability provided by learning from high-quality in-sample data, while also effectively incorporating potential valuable out-of-sample actions. Our algorithm maintains computational efficiency and does not significantly increase the complexity compared to the baseline IQL model.

Experimental results demonstrate that EIQL not only enhances performance but also represents the first application of in-sample offline reinforcement learning as a regularization strategy. This dual learning capability enables our algorithm to mitigate the impact of maximum error extrapolation caused by out-of-sample (OOD) data, achieving robust performance across varied scenarios.

In future work, we plan to explore the potential of our method to improve other in-sample learning techniques, with the aim of assessing the generalizability of our approach across different domains and settings.

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

## A Appendix

Let $Q^*$ represents $\max_{a \in \mathcal{A}, \pi_\beta(a|s) > 0} Q^*(s, a)$ for the simplicity of the notation. Consider the function defined by

$$V_{\tau, \tau'}(s) = (1 - p) \cdot E^\tau_{(s,a) \sim D}[Q_{\tau, \tau'}(s, a)] + p \cdot E^{\tau'}_{s \sim D, a \sim \pi(s)}[Q_{\tau, \tau'}(s, a)],$$

where $p$ is a parameter in the interval [0,1]. It is established in Kostrikov et al. (2021) that

$$V_\tau^I(s) \leq Q^*,$$

where $\pi_\beta$ denotes the behavior policy under which the data $D$ was collected and $Q^*$ represents the optimal action-value function for the decision process.

Firstly, note that at $p = 0$,

$$V_{\tau,\tau'}(s) = E_{(s,a)\sim D}^\tau[Q_\tau(s,a)] \leq Q^*.$$

This initial condition establishes that the function starts below $Q_\tau(s,a)$ at $p = 0$.

The derivative of $V_\tau$, with respect to $p$ is given by

$$\frac{d}{dp}V_{\tau,\tau'}(s) = -E_{(s,a)\sim D}^\tau[Q_{\tau,\tau'}(s,a)] + E_{s\sim D, a\sim\pi(s)}^{\tau'}[Q_{\tau,\tau'}(s,a)].$$

The derivative indicates the rate of change of $V_{\tau,\tau'}(s)$ with respect to $p$. The function $V_{\tau,\tau'}(s)$ will be increasing if $E_{s\sim D, a\sim\pi(s)}^{\tau'}[Q_{\tau,\tau'}(s,a)] > E_{(s,a)\sim D}^\tau[Q_{\tau,\tau'}(s,a)]$ and decreasing if $E_{s\sim D, a\sim\pi(s)}^{\tau'}[Q_{\tau,\tau'}(s,a)] < E_{(s,a)\sim D}^\tau[Q_{\tau,\tau'}(s,a)]$.

We need to establish that there exists a $p > 0$ small enough such that $V_{\tau,\tau'}(s) < Q^*$. Consider two cases based on the value of $E_{s\sim D, a\sim\pi(s)}^{\tau'}[Q_\tau(s,a)]$:

**Case 1:** $E_{s\sim D, a\sim\pi(s)}^{\tau'}[Q_{\tau,\tau'}(s,a)] < E_{(s,a)\sim D}^\tau[Q_{\tau,\tau'}(s,a)]$.

If $E_{s\sim D, a\sim\pi(s)}^{\tau,\tau'}[Q_\tau(s,a)] \leq E_{(s,a)\sim D}^\tau[Q_\tau(s,a)]$, $V_\tau(s)$ is decreasing or constant, and since $V_{\tau,\tau'}(s)(0) < Q^*$, $V_{\tau,\tau'}(s)$ will remain less than $Q^*$.

**Case 2:** $E_{s\sim D, a\sim\pi(s)}^\tau[Q_\tau(s,a)] > Q_\tau(s,a)$

We know:

$$V_\tau(s)(0) = E_{(s,a)\sim D}^\tau[Q_\tau(s,a)] \leq Q^*$$

Given that $V_{\tau,\tau'}(s)$ changes continuously with respect to some parameter $p$, and initially $V_{\tau,\tau'}(s)(0) \leq Q^*$, we aim to show that there exists an interval $[0,\epsilon)$ for some $\epsilon > 0$ where $V_{\tau,\tau'}(s) \leq Q^*$ holds for all $p$ in this interval. To establish this, consider the following reasoning:

1. **Continuity Argument:** By the continuity of $V_{\tau,\tau'}(s)$ over the interval $[0,\epsilon)$ with respect to $p$, and knowing that $V_{\tau,\tau'}(s)(0) \leq Q^*x$, the function $V_{\tau,\tau'}(s)$ remains less than $Q^*$ near $p = 0$.

2. **Application of the Intermediate Value Theorem:** Since $V_{\tau,\tau'}(s)$ is continuous and initially less than $Q^*$, by the Intermediate Value Theorem, if there were any point $p'$ in $[0,\epsilon)$ where $V_{\tau,\tau'}(s)(p') > Q^*$, then there must exist some $p'' \in (0,p')$ such that $V_{\tau,\tau'}(s)(p'') = Q^*$. However, given that $V_{\tau,\tau'}(s)(0) \leq Q^*$ and assuming $V_{\tau,\tau'}(s)$ does not meet or exceed $Q^*$ within the interval $[0,\epsilon)$, the condition $V_{\tau,\tau'}(s) < Q^*$ remains true for all $p$ in this interval.

Thus, it follows that: There exists a $p$ s.t.

$$\forall p \in [0,\epsilon), \ V_{\tau,\tau'}(s)(p) \leq Q^*$$

