# OpenReview forum: "Optimizing Q-Learning Using Expectile Regression: A Dual Approach to Handle In-Sample and Out-of-Sample Data"
_ICLR.cc/2025/Conference — Submitted to ICLR 2025_

### Official Review · Reviewer_7kYU · 2024-10-24

**Soundness:** 2
**Presentation:** 2
**Contribution:** 2
**Rating:** 3
**Confidence:** 4

**Summary:**

The paper proposes an extension of the Implicit Q-Learning (IQL) algorithm by selectively using the maximization capability of Bellman updates controlled using a Bernoulli parameter. Empirical gains are demonstrated with respect to IQL on D4RL offline RL benchmark.

**Strengths:**

1. The paper is well structured. The proposed idea is simple and easy to implement.
2. Empirical results are provided on various robotics benchmarks, showcasing the effectiveness of the approach.

**Weaknesses:**

1. The paper could benefit from a small discussion on model-based approaches for offline RL, such as MOReL[1] and MOPO[2].
Penalizing OOD (Out-of-Distribution) actions is not the only way to handle action extrapolation in offline RL.

2. The related work section is missing key references, particularly the omission of Sparse Q-learning (SQL) and Exponential Q-learning (EQL) [3], which is critical. These methods have outperformed IQL on suboptimal trajectories.

3. The paper does not explain Implicit Q-Learning (IQL) in the preliminaries, making it difficult to follow Equation 1. It is important to describe the notations used with the equation to improve readability.

Minor:

Typo in quotes line 321

No hyperparameters have been provided in appendix making it challenging to reproduce the work

[1] Kidambi, Rahul, et al. "Morel: Model-based offline reinforcement learning." Advances in neural information processing systems 33 (2020): 21810-21823.

[2] Yu, Tianhe, et al. "Mopo: Model-based offline policy optimization." Advances in Neural Information Processing Systems 33 (2020): 14129-14142.

[3] Xu, Haoran, et al. "Offline rl with no ood actions: In-sample learning via implicit value regularization." arXiv preprint arXiv:2303.15810 (2023).

**Questions:**

Q1. Why is the Bernoulli parameter required during policy extraction in Eq 3? How is a selected in second part of the equation where B' is used?

Q2. I am confused about Theorem 4.1. The notations seem inconsistent. In Eq1 B is used  in theorem p is use. The proof states "This proof highlights
the influence of the parameter $\beta$ in controlling the extent to which the new policy deviates from optimality", However $\beta$ as per my understanding denotes the behavior policy and is nota parameter?
If the authors could provide clear explanation about each step and a notation table the contributions will be better understood.

Q3 What was the choice of B for each environment during empirical evaluation? Why does EIQL have lower performance than some baselines in Table 1?

Kindly also refer to the weaknesses.

---

### Official Review · Reviewer_pBMH · 2024-10-28

**Soundness:** 1
**Presentation:** 2
**Contribution:** 1
**Rating:** 1
**Confidence:** 4

**Summary:**

This paper proposes two approaches to extend Implicit Q-Learning: a) it modifies the value loss by incorporating a target of sampled action value function, and the policy objective by sampled advantage weighted regression.

**Strengths:**

No

**Weaknesses:**

1. The theoretical analysis is misleading as a loss function cannot be regarded as a random variable.
2. The experiments exclude the SOTA offline RL algorithms

**Questions:**

1. Where is the definition of $L^\tau_2$?
2. Where does the action in the second part of Equation 3 come from?

---

### Official Review · Reviewer_X8fM · 2024-11-03

**Soundness:** 1
**Presentation:** 1
**Contribution:** 2
**Rating:** 3
**Confidence:** 3

**Summary:**

In this work, the authors propose to modify the value loss in Implicit Q-Learning to stochastically tradeoff between using in-sample data to estimate the loss and using the learned policy to sample potentially unseen actions.

**Strengths:**

The problem being worked on is well-motivated and important. There are a lot of experiments with potentially significant results.

**Weaknesses:**

I had a hard time understanding this paper. The writing is unclear in many parts and there are inconsistencies in the notation. I detail some of them below, but I think the amount of rewriting required to get this paper to an acceptable level is beyond what can be done in the rebuttal period.

- I understand that this work builds on prior works, however, every paper should stand on its own in some way. In this work, the authors build on Implicit Q-Learning, but there have been no introductions of IQL even just to set up the notation, the background and give some context. It is not necessary to give an in-depth discussion but if it is a central work that is being built upon, a short summary or setup would clarify what is being added.
  - For example, $L_2^\tau$ is not defined anywhere in the current work. Similarly, L164 mentions that the value function in (Kostrikov et al., 2021) is about to be defined but there is no context as to why, what it means, or how it relates to the current work. The few words in L171-173 only add to the confusion as it seems to assume that the reader is coming in with the discussion of (Kostrikov et al, 2021) fresh in their minds.
- The notation changes from L163 to L167-170 for the $\tau$-th expectile of a random variable $X$.

- Some examples should be cited in L097 for the offline RL algorithms that are alluded to.
- It’s unclear what Section 4.3 is trying to show until the end of the section, this should be stated clearly at the beginning.

- 3 seeds for the experimental results is too small. Even just comparing to IQL, there should be at least 10 per experiment.
- Figures 1 and 2 looks like each experiment was only done with one seed, which makes it hard to draw conclusions with confidence. I also don’t see how the conclusion in L358-359 was derived from Figure 2. All the methods look pretty much the same.

Minor:
- The citations should be in parenthesis when they are not used as the subject of the sentence (e.g. in 042, it should be (Levin et al, 2020), similarly in L045, L048, etc.
- Eq (2), $V_\psi$ has not rendered correctly
- The fonts and rendering of Figures 1 and 2 are too small to be readable on paper and become blurry when zoomed in
- These figures are labelled as “analysis” but I would call them experimental results.

**Questions:**

- Is $\tau’$ related to $\tau$ or are they meant to be independent parameters?
- In L080, $p$ was used to denote the transition dynamics, is that still the case for the $p$ in L108-L114?
- What is $\beta$ in Theorem 4.1?

---

### Official Review · Reviewer_JGTd · 2024-11-04

**Soundness:** 1
**Presentation:** 1
**Contribution:** 1
**Rating:** 1
**Confidence:** 5

**Summary:**

This paper presents Extended Implicit Q-Learning (EIQL), a new approach to offline reinforcement learning that aims to extend Implicit Q-Learning by selecting actions beyond the dataset constraints to leverage the Bellman update's maximization capability. The proposed method intends to improve performance in environments with sparse rewards or incomplete trajectories by occasionally incorporating actions not seen in the dataset. Experimental evaluations are conducted on standard offline RL benchmarks to illustrate EIQL’s potential benefits over traditional algorithms.

**Strengths:**

N/A

**Weaknesses:**

This paper exhibits a lack of rigor and completeness, with several critical issues that impact both readability and credibility. The related work section is notably underdeveloped, which severely hampers an understanding of how this work fits within the existing body of research and fails to establish a clear contribution. Additionally, the paper's notation is inconsistent and ambiguous, with numerous undefined terms and unclear derivations, leading to confusion in understanding the methodology. Key elements such as Theorem 4.1 are poorly presented, lacking in both formal results and coherence. Section 4.3 appears disconnected from the rest of the paper, as there is minimal context or explanation for its inclusion. Moreover, the paper is marred by formatting errors, including unformatted algorithm environments and incorrect citation formats, which contribute to an unprofessional presentation. In its current form, the paper does not meet the standards expected for an ICLR submission and would benefit greatly from a substantial revision before being reconsidered for publication.

**Questions:**

.

---

### Meta-Review · Area_Chair_EnPQ · 2024-12-19

**Metareview:**

Optimizing Q-Learning Using Expectile Regression: A Dual Approach to Handle In-Sample and Out-of-Sample Data

Summary: This paper proposes an algorithm for offline RL, by suggesting an extension to an existing work (Implicit Q-Learning (Kostrikov et al. 2021)). The paper presents simulation experiment results performed on various mujoco tasks.

Comment: This paper received four expert reviews, with scores 1, 1, 3, 3, and the average score is 2. All reviewers pointed out many issues, starting from the poor quality of presentation to the lack of algorithmic novelty and poor experiments.  Based on the reviewers, this paper is currently below the acceptance quality for a top ML conference.

**Additional Comments On Reviewer Discussion:**

The authors have not provided a response.

---

### Decision · Program_Chairs · 2025-01-22

Reject